# OpenReview forum: "Privacy-Aware Visual Language Models"
_TMLR — Accepted by TMLR_

### Review · Reviewer_kM4i · 2026-02-28

**Summary Of Contributions:**

This paper studies privacy-awareness in Visual Language Models (VLMs), identifying a gap between strong multimodal reasoning performance and the ability to recognize privacy-sensitive visual content. The authors first analyze existing privacy datasets and empirically demonstrate substantial label inconsistencies. They then introduce two compact and manually curated evaluation benchmarks, PrivBench and PrivBench-H, aligned with GDPR-style privacy categories. In addition, they propose PrivTune, an instruction-tuning dataset designed to improve privacy sensitivity in VLMs.

A key empirical finding is that fine-tuning a lightweight VLM (TinyLLaVA) on as few as 100 privacy-focused samples significantly improves privacy recognition (up to 0.86 MCC on PrivBench), while only slightly degrading performance on standard benchmarks. The paper further evaluates generalization to held-out privacy categories and applies the privacy-tuned model to large-scale dataset auditing (Places365).

Strengths

1. Addresses an underexplored yet societally critical problem: visual privacy awareness in VLMs.

2. Careful qualitative and quantitative analysis of label noise in existing privacy datasets.

3. Demonstrates strong gains from extremely small-scale fine-tuning.

4. Includes generalization, multilingual, and real-world dataset auditing experiments.

Weaknesses

1. Privacy definition is somewhat narrow and heavily object-centric.

2. Evaluation task reduces privacy to binary detection (“contains private object or not”), limiting conceptual depth.

3. Training data relies partly on GPT-4-generated dialogues, raising circularity concerns.

4. Some claims (e.g., generalization and safety implications) are stronger than experimental evidence fully supports.

**Audience:**

Yes

**Audience Explanation:**

The intersection of multimodal models, dataset quality, safety alignment, and regulatory concerns (e.g., GDPR) is highly relevant to TMLR’s audience.

**Claims And Evidence:**

Yes

**Claims Explanation:**

The empirical claims regarding improved MCC performance after privacy-tuning are clearly supported by extensive benchmarking across PrivBench, PrivBench-H, VISPR, and additional ablations. The experiments are reproducible in principle and include cross-validation analysis and data-efficiency studies.

However, several claims could be tempered:

1. Generalization beyond trained privacy categories
While leave-one-class-out experiments are promising, the taxonomy remains limited to object-level privacy cues (e.g., passport, credit card, face, license plate). This does not fully demonstrate conceptual generalization to contextual or relational privacy risks.

2. Misalignment between vision and text spaces (GPT-4 claim)
The discussion of GPT-4 "misalignment" is interesting but speculative. The paper shows performance discrepancies but does not isolate whether this is due to policy-based refusals, vision encoder bias, or prompt framing.

3. Societal privacy interpretation (multilingual results)
The attribution of improved German performance to cultural privacy norms is not substantiated experimentally and should be framed as hypothesis rather than interpretation.

Overall, the empirical evidence supports the main technical claims, but some broader interpretations would benefit from more rigorous validation or toned-down phrasing.

**Requested Changes:**

See above

---

> ### Author Response · Authors · 2026-04-10
> **Official Comment by Authors**
>
> We thank the reviewer for the careful and constructive reading of our work.
>
> **Privacy Definition**
>
> We appreciate this observation and want to clarify that the scope of our taxonomy is a deliberate choice rather than a limitation. Our classes are explicitly aligned with GDPR Articles 4 and 9, which define personal data as information that is directly traceable to a specific individual. Passports, faces, fingerprints, license plates, and similar identifiers fall unambiguously into this category across cultural and legal contexts.
>
> We agree that privacy is a broader concept than our taxonomy captures. More contextual attributes, such as hair colour or handwriting, can be private under specific circumstances, but their privacy status is more ambiguous and context-dependent. The narrow scope is therefore a feature; it enables clean, reliable measurement of privacy awareness.
>
> **Binary Evaluation**
>
> A central goal of our work is to measure whether VLMs understand what constitutes private visual content; do they recognise a passport or a fingerprint as private at all? Binary classification directly operationalises this question in a way that is both reproducible and comparable across models.
>
> *Graded scoring was attempted but regressed to binary signals.* As noted in Section 5, we experimented with privacy scores ranging from 1 to 5 to capture gradations in privacy levels. However, we found that all evaluated VLMs consistently collapsed into binary responses, showing no variance across the scale.
>
> *Open-ended evaluation introduces its own serious challenges.* An open-ended evaluation protocol is appealing in principle, but would require reliable automated evaluation of privacy reasoning. A growing body of work documents the unreliability of LLM-as-a-Judge [1], with low human expert agreement rates in knowledge-intensive fields [2]. Privacy is precisely such a domain: legally defined, and requiring specialised judgment. Crucially, our paper demonstrates that state-of-the-art VLMs misunderstand visual privacy; using these same models to judge open-ended privacy reasoning could therefore be unreliable.
>
> *The paper already goes beyond binary detection.* Our PrivTune dataset and the resulting Privacy VLMs operate in a non-binary output space. Furthermore, the Places365 application in Section 5.1 produces privacy predictions across location categories alongside natural language explanations, with GPT-4 analysis confirming the model reasons about contextual sensitivity in medical, and military settings.
>
> [1] Gu et al. (2024). A survey on LLM-as-a-Judge. The Innovation.
>
> [2] Szymanski et al. (2025). Limitations of the LLM-as-a-Judge approach for evaluating LLM outputs in expert knowledge tasks. IUI 2025.
>
> **GPT-4 Circularity**
>
> GPT-4 is used as a natural language generation tool in PrivTune. It generates conversational dialogues conditioned on images, class names, and privacy labels. It plays no role whatsoever in assigning privacy labels. As formalised in Section 4, the label is always determined by human annotators. GPT-4 never decides whether an image is private or public.
>
> There is therefore no circularity in the evaluation. We fine-tune small-scale VLMs such as TinyLLaVA and InternVL2.5 on PrivTune and evaluate the resulting Privacy VLMs on PrivBench and VISPR. These models outperform GPT-4, which would be impossible if the models were mimicking GPT-4's privacy judgements.
>
> **Claims**
>
> We have revised the relevant passages accordingly.

---

### Review · Reviewer_HbeM · 2026-03-05

**Summary Of Contributions:**

The paper makes several key contributions to the underexplored area of privacy awareness in Visual Language Models (VLMs):

* **Evaluation of Existing Datasets and Models:** The authors evaluate 10 state-of-the-art VLMs and demonstrate their limitations in identifying visual privacy. Furthermore, they conduct human evaluations to reveal significant label noise and inconsistencies in existing visual privacy datasets like Biv-Priv and PrivAlert.

* **New Benchmarks:** The authors introduce two high-quality, human-curated benchmarks, PRIVBENCH and PRIVBENCH-H (a harder variant with challenging negative samples). These are explicitly mapped to commonly recognized privacy categories under the GDPR.

* **Instruction-Tuning Dataset and Privacy VLM:** The paper introduces PRIVTUNE, a dataset of multi-turn privacy-aware dialogues generated via GPT-4 and ShareGPT.

* **Efficient Privacy-Tuning:** The authors demonstrate that fine-tuning an off-the-shelf VLM (TinyLLaVA) with just 100 samples from PRIVTUNE produces a "Privacy VLM" that substantially outperforms state-of-the-art models (including GPT-4) on privacy recognition tasks. This is achieved with minimal degradation on standard VLM benchmarks.

**Additional Comments:**

N/A.

**Audience:**

Yes

**Audience Explanation:**

Yes. This paper studies an underexplored aspect of multimodal safety: visual privacy awareness in VLMs, and introduces new benchmarks and datasets that provide useful evaluation tools and empirical insights likely to interest researchers working on multimodal models, safety, and evaluation.

**Broader Impact Concerns:**

N/A.

**Claims And Evidence:**

Yes

**Claims Explanation:**

The authors provide comprehensive empirical evidence to support their core claims:

- **Dataset Noise:** The claim that existing datasets suffer from label noise is backed by a human evaluation study measuring binary accuracy and Fleiss' Kappa, which clearly shows their PRIVBENCH dataset is far more consistent than prior work. Qualitative examples of mislabeling (e.g., black screens labeled as private) are also provided.

- **VLM Limitations:** The lack of privacy awareness in current VLMs is thoroughly supported by zero-shot evaluations across multiple models using the Matthews Correlation Coefficient (MCC).

- **Efficacy and Efficiency of Privacy-Tuning:** The authors provide strong evidence that their tuning method works. Table 2 shows the Privacy VLM achieving an MCC of 0.86 on PRIVBENCH, far exceeding baselines. An ablation study on data volume (Figure 6) convincingly supports the claim that only ~100 samples are needed.

* **Generalization:** The claim that the model can generalize to unseen private categories is supported by a leave-one-out cross-validation experiment (Table 4) where the model still performs well on a privacy class excluded from its training.

**Requested Changes:**

To strengthen the paper before publication, I recommend the following adjustments:

1. **Generalization Across Base Models:** The authors successfully fine-tuned a TinyLLaVA model. To prove the versatility of the PRIVTUNE dataset, it would be highly beneficial to apply the same 100-sample fine-tuning to at least one other VLM architecture (e.g., standard LLaVA or CogVLM) and report if the gains are consistent.

2. **Prompt Sensitivity Transparency:** The methodology mentions that the authors tested prompt variations using PRIVTUNE and selected the best-performing prompt per model for the final PRIVBENCH evaluation. To ensure reproducibility, the exact prompts used for each baseline model must be explicitly listed in the Appendix.

3. **Data Contamination Discussion:** Since the datasets were curated from Re-LAION-5B, the authors should briefly discuss the likelihood of data contamination.

---

> ### Author Response · Authors · 2026-04-10
> **Official Comment by Authors**
>
> We thank the reviewer for the constructive feedback.
>
> **Generalisation Across Base Models**
>
> To demonstrate that privacy-tuning generalises beyond a single architecture, we fine-tuned two models from the InternVL2.5 family, InternVL2.5-2B (InternLM2-1.8B backbone) and InternVL2.5-4B (Phi-3-mini-128k backbone), on PrivTune. These models use a different vision encoder (InternViT-300M) and different language model backbones from TinyLLaVA (Phi-2, SigLIP), providing a meaningful architectural contrast. Both models show substantial and consistent improvements after privacy-tuning, with minimal degradation on standard benchmarks (VQAv2, POPE, ScienceQA). Results are included in the main text, and hyperparameters are provided in the Appendix.
>
> **Prompt Sensitivity Transparency**
>
> *Note on Revision: During the preparation of this revision, we identified and corrected an inconsistency in our prompt selection procedure; the analysis was initially run under an earlier experimental setup. A small number of model scores have changed slightly as a result, but the main results and conclusions of the paper are unaffected.*
>
> All prompts used per model are now documented in Appendix C, including a prompt sensitivity analysis.
>
>
> **Data Contamination Discussion**
>
> Some evaluated VLMs may have been pre-trained on data derived from LAION-5B and could therefore have encountered individual images before. However, the privacy-specific image-text pairs in our benchmarks are entirely new. We have added a brief discussion of this in the Discussion section.

---

### Review · Reviewer_EPNE · 2026-03-28

**Summary Of Contributions:**

This work introduces two benchmarks, PrivBench and PrivBench-H, for VLMs to access their privacy awareness, as well as PrivTune, a dataset to tune VLMs and improve their privacy awareness. A key empirical finding is that several existing privacy datasets are too noisy to serve as reliable benchmarks, and the authors document substantial label inconsistencies in BivPriv and PrivAlert, while also noting that VISPR, though higher quality, includes more debatable privacy classes and is heavily skewed toward images containing people.
Using PrivTune, they privacy-tune a TinyLLaVA-based model with LoRA and show that even fine-tuning on only 100 images can produce large gains, outperforming strong zero-shot VLM baselines and even GPT-4 on their privacy benchmarks while only slightly degrading performance.

---

*Weakness*

- I personally thought 160 might not be a large one, so the paper's broader claims about robust real-world privacy generalization feel stronger than what the scale of the evidence fully supports.
- it is unclear whether the gains would transfer similarly across stronger or more diverse VLM families, as the training is done only over LoRA with one base model.

**Additional Comments:**

N/A

**Audience:**

Yes

**Audience Explanation:**

Yes.

The dataset can be useful for both training and evaluation, which is pretty good for the community to assess the privacy VLM capabilities. Also, note that current VLM datasets regarding privacy isn't that much, which is a quite timely contribution.

**Broader Impact Concerns:**

Since this is a dataset work, you should provide the licenses to those images in your main text. The point in Appendix A.1 is not enough. Also, the proposed datasets contain sensitive personal images and are derived from re-laion-5b, the new version should clearly specify the licensing status of the source images, whether redistribution is permitted, and what permission or legal basis supports inclusion of private images in the released benchmark. Note that in other venues, such as NeurIPS D&B, without specifying these details will usually incur ethical inspections.

**Claims And Evidence:**

Yes

**Claims Explanation:**

Yes.

I found the benchmark design pretty clear (with rationale explained in the main text as well as further motivation explanation in the appendix, which I suggest to move the appx to the main text, see my requested for changes), also with human evaluation of label quality, and strong privacy-tuning results, though my concern is that the evidence might be less convincing for broad real-world generalization because the benchmarks are small and centered on a binary privacy setup.

**Requested Changes:**

- Move the appendix after the main text, for easier reading.
- Ethical concerns, see broader impact concerns section.
- Can you add more experiments over finetuning to justify that the training results can be generalized to more model families?

---

> ### Author Response · Authors · 2026-04-10
> **Official Comment by Authors**
>
> We thank the reviewer for the constructive feedback.
>
> **Privacy-Tuning on other VLMs**
>
> To validate that privacy-tuning generalises beyond TinyLLaVA, we additionally fine-tuned InternVL2.5-2B and InternVL2.5-4B on PrivTune using the same procedure. These models differ substantially from TinyLLaVA in both architecture and scale, using InternViT-300M as the vision encoder and InternLM2-1.8B or Phi-3-mini-128k as the language model backbone. Both models show consistent and substantial improvements after privacy-tuning, with only minimal degradation on standard benchmarks, confirming that the gains are not specific to a single model family. Updated results are included in the main text, and hyperparameters are provided in the Appendix.
>
> **Generalisation Claim**
>
> We have revised the generalisation discussion to more precisely scope what the evidence supports.
>
> **Dataset Licensing**
>
> We have added the licensing to Section 4 and updated the datasheet in Appendix A accordingly.
>
> Our datasets consist of the privacy labels, dataset splits, and PrivTune dialogues, which are released under CC BY-NC 4.0. Regarding the images, we do not redistribute them; we share only the URLs pointing to the original publicly available sources in Re-LAION-5B, consistent with how LAION itself operates. Researchers who request access agree to delete all downloaded images after use.
>
> **Appendix after Main Text**
>
> We put the motivation for the datasets in the Methodology Section.

---

> > ### Comment · Reviewer_EPNE · 2026-04-13
> >
> > Thanks for the response. I think main concern regarding the generalization & diverse model family is addressed. I still suggest to move the appendix directly after the main text as some experiment results and explanation could be better re-organized with the main text to improve the readability. Different from CVPR/ICCV/WACV, TMLR encourages and allows a detailed technical appendix just after the main text to make the readability better.
> >
> > I think this dataset overall is a interesting and timely contribution to the community, and I am happy to submit my recommendation of acceptance.

---

> > > ### Author Response · Authors · 2026-04-24
> > > **Official Comment by Authors**
> > >
> > > We thank the reviewer for the suggestion and have uploaded a revised version with the appendix after the main text.

---

### Decision · Action_Editor_CSNT · 2026-05-21

**Recommendation:** Accept as is

**Audience:**

Yes

**Audience Explanation:**

The topic and problem addressed by the paper are timely. Although there remains room for improvement, all reviewers and I agree that the material developed in the paper should be of interest to the community.

**Claims And Evidence:**

Yes

**Claims Explanation:**

The paper addresses the problem of privacy awareness in Vision-Language Models (VLMs). The authors first analyze existing privacy datasets (BivPriv, PrivAlert, VISPR) and empirically document label inconsistencies. They then introduce two manually curated evaluation benchmarks aligned with GDPR-style privacy categories, as well as an instruction-tuning dataset designed to improve privacy sensitivity in VLMs. Using this dataset, they fine-tune a TinyLLaVA-based model with LoRA and show that training on 100 privacy-focused samples yields gains in privacy recognition, outperforming strong zero-shot VLM baselines and GPT-4 on their benchmarks, while only slightly degrading performance on standard tasks. The paper also evaluates generalization to held‑out categories and applies the privacy‑tuned model to large‑scale dataset auditing.